# Evolutionary Neural Architecture Search with Performance Predictor Based on Hybrid Encodings

Jiamin Xiao
*School of Automation*
*Guangdong University of Technology*
Guangzhou 510006, China
2112204145@mail2.gdut.edu.cn

Kuoyong Yu
*Beijing National Innovation*
*Institute of Lightweight Ltd.*
Beijing 100875, China

Bo Zhao*
*School of Systems Science*
*Beijing Normal University*
Beijing 100875, China
zhaobo@bnu.edu.cn

Derong Liu
*Department of Mechanical and Energy Engineering*, *Southern University of Science and Technology*
Shenzhen 518055, China, liudr@sustech.edu.cn
*Department of Electrical and Computer Engineering*, *University of Illinois Chicago*
Chicago, IL 60607, USA, derong@uic.edu

*Abstract*—Neural architecture search (NAS) has received a lot of attention since the development of deep neural networks (DNNs) in various scientific and application fields. By learning the relationship between neural network architectures and their corresponding performance, the performance predictor which plays a critical role in the NAS methods exactly improves the efficiency. However, the efficiency of performace predictors mainly depends on the training approaches of performance predictors and the encoding approches of neural network architectures. In this paper, we propose a hybrid encoding-based predictor building upon two computation-aware encodings with different training approaches. It trains a generative module by unsupervised learning to better encode architectures and a graph flow module by supervised learning to reduce the cost of evaluated architectures, which are beneficial to the search for the optimal architecture representation in the latent space. Additionally, an evolutionary neural architecture search method (HEP-ENAS) is proposed to efficiently explore the promising architectures by applying the hybrid encoding-based performance predictor to the covariance matrix adaptation evolution strategy (CMA-ES). A series of experiments conducted on NAS-Benmarks demonstrate the benefits of hybrid encoding-based predictor for searching for the optimal architecture in the latent space and the effectiveness of HEP-ENAS compared with popular NAS methods.

*Index Terms*—Neural architecture search, evolutionary algorithm, predictor-based search, deep generative model, hybrid encodings.

## I. INTRODUCTION

Due to the versatile capabilities of automatic neural network design, neural architecture search (NAS) has widely applied to many scientific and application fields such as image classification, object detection, and natural language processing [1]–[3]. However, NAS is time-consuming if it has to train and evaluate numerous architectures to acquire their absolute performance. More critically, the occasional incorrect

This work was supported in part by the National Natural Science Foundation of China under grants 62073085, 61973330 and 62350055, in part by the Shenzhen Science and Technology Program under grant JCYJ20230807093513027, in part by the Fundamental Research Funds for the Central Universities under grant 1243300008, and in part by the Beijing Normal University Tang Scholar. *(Corresponding author: Bo Zhao)*

orientation of traditional search algorithms also reduce the efficiency of NAS methods. To overcome these challenges, performance predictors are proposed to learn the relationship between neural architectures and corresponding performance indicators, so that the performance indicators of neural architectures can be acquired by predictors without training the neural architectures from scratch. It has been demonstrated that the performance predictor plays a critical role in NAS to accelerate the search process and improve the performance of final searched architectures.

Considerable researches have developed various predictors to guide the search process into the promising area or estimate the performance of candidate architectures. The efficiency of these predictors greatly depends on the training approaches of performance predictors and the encoding methods of neural network architectures. The training approaches of performance predictors are mainly divided to two types, i.e., training-free approaches and training-based approaches.

Training-free predictors such as ProxyBO [4] aim to predict the performance of neural architectures without training process. However, training-free predictors usually perform well on some certain architectures only. Training-based predictors commonly consist of an encoder module and a regressor module. The encoder module maps the neural architectures into architectural feature encodings, and the regressor module predicts the performance of neural architectures based on the architectural feature encodings. Once the training process is completed, training-based predictors can efficiently predict the scores of candidate neural architectures, and significantly accelerate the search process of NAS. Additionally, CTNAS [5] has shown that the ranking of candidate architectures based on the prediction scores is comparatively easier than predicting precise performance indicators of candidate architectures in the predictor-based NAS.

Encoding methods are particularly important in the prediction-based NAS due to its great impact on the effectiveness of performance predictors. The encodings of neural

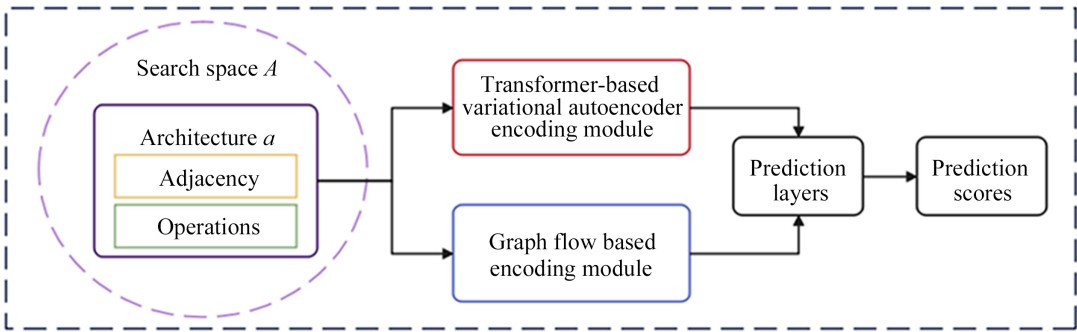

Fig. 1. The overview of the proposed hybrid encoding-based performance predictor.

network architectures can be mainly divided to four categories, i.e., structure-aware encodings, score-aware encodings, computation-aware encodings and others. structure-aware encodings commonly treat the architectures as the directed acyclic graphs (DAGs) represented by the adjacency and operation matrices. Some structure-aware encodings represent architectures by converting the adjacency and operation matrices into special vectors. Adjacency encodings and path encodings are two typical structure-aware encodings [6]. The adjacency encodings concatenate the adjacency matrix with a list of operation labels, or a list of indices of possible edges (the flattened adjacency matrix) with the list of operation labels to represent architectures. The path encodings utilize a set of paths from input node to output node within the architecture DAGs to represent architectures, and each path is denoted by a one-hot vector or a categorical feature. Score-aware encodings use a vector to measure the indicators of architectures such as activations, gradients and properties. Zero-cost proxies (ZCPs) [4], which is a common type of score-aware encodings, is usually used to predict the accuracy of neural architectures. Since its easily accessible characteristics, zero-cost proxies usually are applied in training-free predictors. Its goal is to find the relationship between the architectural features and the architectural performance indicators. Although there are no explicit structure information of neural architectures in zero-cost proxies, zero-cost proxies still implicitly contain the related architectural properties. Computation-aware encodings aim to distill the the architectural structure information related to the performance indicators of architectures through the computation. There are two types of computation-aware encodings, i.e., unsupervised computation-aware encodings and supervised computation-aware encodings. The unsupervised computation-aware encodings such as Arch2Vec [14] and D-VAE [8] map the architectures with different structure information but similar performance into the same latent space region. In addition, the smoothness and continuity of latent space allow the unsupervised computation-aware encodings improve the efficiency of the downstream search optimization algorithms. Different to the unsupervised ones, supervised computation-aware encodings map the architectures into the latent space in a supervised learning manner, and they are

continually evolved by sampling new architecture-accuracy pairs. Owing to the labeled property of supervised learning, supervised computation-aware encodings exhibit the high prediction performance, but more likely to be limited in the specific task they are trained. It means the extendibility and transferability of the predictors using supervised computation-aware encodings cannot always perform well.

The experiments in FLAN [9] have shown that supervised computation-aware encodings often out-perform other encoding methods. However, the supervised training process is expensive due to it requires to train numerous neural architectures, and it is not always feasible to acquire the architectural performance indicators of sufficient architectures. Compared to the supervised computation-aware encodings, unsupervised computation-aware encodings are able to save a significant amount of computational resources while capturing underlying architectural information. In this paper, we propose a novel hybrid encoding-based performance predictor to improve the sample efficiency of predictors, which combines the supervised computation-aware encodings with the unsupervised computation-aware encodings together. Their combination accelerates the search process in NAS and improves prediction accuracy of predictors. Furthermore, the hybrid encoding-based performance predictor is applied to the evolutionary optimization algorithm to propose a novel evolutionary architecture search algorithm, i.e., evolutionary architecture search algorithm with hybrid encoding-based predictor (HEP-ENAS). The contributions of the paper can be concluded as follows.

- The hybrid encoding-based performance predictor is developed to better balance the budget between efficiency and performance of predictors. To demonstrate the benefits, we propose the HEP-ENAS algorithm, which integrates with the hybrid encoding-based performance predictor to CMA-ES algorithm to search for the promising architectures in the continuous latent space.
- The experimental results conducted on the NAS-Bench-101 and NAS-Bench-201 benchmarks demonstrate the effectiveness of HEP-ENAS. Particularly, HEP-ENAS achieves better performance than several NAS with different encoding-based predictors.

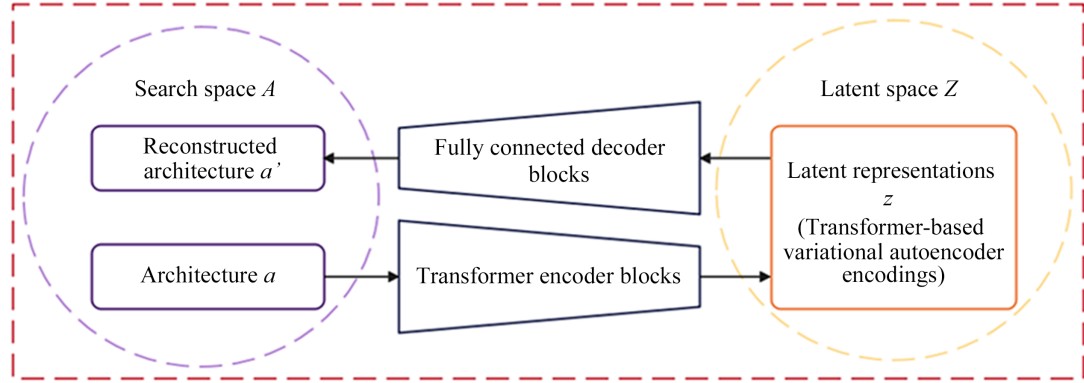

Fig. 2. The framework of the Transformer-based variational autoencoder encoding module.

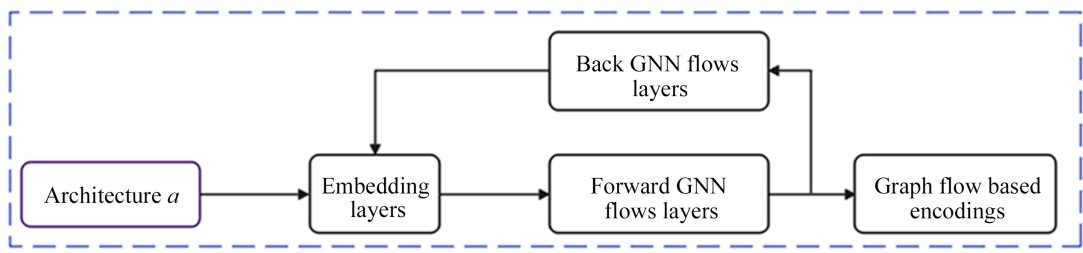

Fig. 3. The framework of the Graph flow-based encoding module.

## II. THE HEP-ENAS ALGORITHM

NAS aims to automatically design the optimal neural architecture with limited resources. This search process can be formulated as a bi-level optimization problem

$$
\begin{aligned}
\max_{\alpha \in A} \quad & \text{Validation.accuracy}(W^*(\alpha), \alpha, \mathcal{D}_{\text{val}}) \\
\text{s.t.} \quad & W^*(\alpha) = \underset{W(\alpha)}{\operatorname{argmin}}\, \mathcal{L}_{\text{train}}(W(\alpha), \alpha, \mathcal{D}_{\text{train}}),
\end{aligned}
\tag{1}
$$

where $\mathcal{L}_{\text{train}}$ is the training loss, $W(\alpha)$ represents the weights of architecture $\alpha$. The objective of architecture search problem is to find the optimal architecture $\alpha^*$ that maximizes the validation accuracy on the validation dataset $\mathcal{D}_{\text{val}}$, where the weights $W^*(\alpha)$ associated with the architecture $\alpha$ are obtained by minimizing the training loss $\mathcal{L}_{\text{train}}$ on the training dataset $\mathcal{D}_{\text{train}}$.

Based on (1), NAS contains three important components, i.e., search space, search strategy and performance evaluation strategy. Moreover, depending on different optimization algorithms in the search strategy, NAS also can be divided into several types, such as Bayesian optimization (BO) methods, reinforcement learning (RL) methods, and evolutionary algorithm (EA) methods. Compared to the BO and RL methods, the candidate neural architectures are generated from the mutations and re-combinations of sampled neural architectures in the EA-based NAS, and the promising architectures with better performance can be generated in the following regressions. In this papar, HEP-ENAS, which is a novel EA-based NAS algorithm , is developed with a hybrid encoding-based performance predictor. The search process of the proposed HEP-ENAS is shown in Algorithm 1.

### A. Hybrid Encoding-based Performance Predictor

As shown in Fig. 1, the proposed hybrid encoding-based performance predictor is built on the basis of a Transformer-based variational autoencoder (VAE) encoding module and a graph flow-based encoding module, which are two diferent computation-aware encoding modules.

As a deep generative model, Transformer-based variational autoencoder encoding module utilizes the Transformer as an encoder to map the neural architectures into the continuous latent space. The performance of computation-aware encodings mainly depends on the performance of neural architecture encoders. Traditional neural architecture encoders, such as multilayer perceptron (MLP), long short-term memory (LSTM) and graph neural network (GNN), limit the architecture representation ability and computation-aware encoding effectiveness. Transformer can be used to train a performance predictor by supervised learning, due to its capability of feature extraction for graph structure data. On the other hand, Transformer captures the locality information to benefit encoding the architectures with simliar performance into the same region. Transformer-based variational autoencoder encoding module is trained using unsupervised learning to map the architecture into the latent space and reconstructs the architectures from latent representations. The generative model learns a continuous representation $z \in Z$ by approximating the posterior distribution, which is achieved by the Kullback-Leibler (KL) divergence between the approximate posterior

**Algorithm 1:** The implement of HEP-ENAS
___

**Input:** Search space $A$, the number of initial samples $N$, maximum evaluated number $M$, the initial parameters of CMA-ES optimizer $W_c$, the parameters of pretrianed hybrid encoding-based predictor $W_p$.

1   $D \leftarrow \textbf{\textit{Randomsample}}(A, N)$, $F \leftarrow \textbf{\textit{Evaluate}}(D)$, $D_z \leftarrow \textbf{\textit{Encode}}(D)$.

2   //Random sampling to acquire the initial architectures with the corresponding performance and latent representations.

3   **while** $|D| < M$ **do**

4      $D_{zt} \leftarrow \textbf{\textit{CMA-ES}}(D_z, W_c)$;

5      //Sampling the candidate architectures.

6      $S_{zt} \leftarrow \textbf{\textit{Predictscores}}(D_{zt}, W_p)$;

7      //Predicting the performance scores of the candidate architectures.

8      $D_{zn} \leftarrow \textbf{\textit{Topk}}(S_{zt}, D_{zt})$;

9      $D_n \leftarrow \textbf{\textit{Decode}}(D_{zn})$;

10     //Selecting the top-$k$ latent representations and decode them to the corresponding architectures.

11     $F_n \leftarrow \textbf{\textit{Evaluate}}(D_n)$;

12     //Evaluating the architectures to obtain performance.

13     $D = D \cup D_n, D_z = D_z \cup D_{zn}, F = F \cup F_n$;

14     $W_c \leftarrow \textbf{\textit{UpdateCMA-ES}}(D_{zt}, S_{zt})$;

15     $W_p \leftarrow \textbf{\textit{Updatepredictor}}(D, D_z, F)$;

16     //Updating the parameters.

17 **end**

**Output:** The optimal neural architecture with the corresponding performance indicators.
___

and the prior distribution. Thus, the training loss is composed of the architecture reconstruction loss and the Kullback-Leibler divergence.

Graph flow-based encoding module inspired by GATES [10] and TA-GATES [11] utilizes the graph-flow mechanisms to capture the local information of architectures. The Transformer-based variational autoencoder encodings implicitly contain the information of architectural context, but may result in the loss of partial architectural topology information. The calculation step by step according to the nodes of graph flow-based encodings exactly compensates the loss of architectural information. To encode the architectures, graph flow-based encodings conduct an iterative process of forward and backward flows for several times. In each iteration, graph flow-based encodings updates the embeddings of architectures based on the backward flows. At the final iteration, the forward flows will be the outputs of graph flow-based encoding module.

Considering an architecture $\alpha \in A$, where $A$ is search space, the computation process of Transformer-based variational autoencoder encoding module can be described as

$$Q_{li}, K_{li}, V_{li} = \text{Embedding}_1(\alpha), \qquad (2)$$

$$H_{li} = \text{softmax}(Q_{li}, K_{li}, V_{li}), \qquad (3)$$

$$\hat{H}_l = \text{Concat}(H_{li}), \qquad (4)$$

$$H_l = \text{ReLU}(\hat{H}_l, W, b), \qquad (5)$$

where the $Q_{li}$, $K_{li}$ and $V_{li}$ represent the query, key and value matrices at $l$th Transformer encoder layer of $i$th attention head, respectively, $H_{li}$, $H_l$ and $\hat{H}_l$ represent the hidden state, $W$ is weight matrix and $b$ is bias matrix, the hidden states of

final Transformer encoder layer are latent representations $z$, which are the outputs of the Transformer-based variational autoencoder encoding module.

The computation process of graph flow-based encoding module can be described as

$$E_0 = \text{Embedding}_2(\alpha), \qquad (6)$$

$$F_I[1] = E_I, \qquad (7)$$

$$F_I[2:N] = \text{Propagation}(F_I[1], E_{I-1}, \alpha), \qquad (8)$$

$$B_I[N] = \text{Feedback}(F_I[N]), \qquad (9)$$

$$B_I[1:N-1] = \text{Propagation}(B_I[N], E_{I-1}, \alpha), \qquad (10)$$

$$E_I = \text{Update}(E_{I-1}, F_I, B_I), \qquad (11)$$

where $E_I$ represents the embeddings of architecture $\alpha$ at $I$th iteration, $F_I$ and $B_I$ represents the forward flows and the backward flows, respectively, $N$ is the number of nodes in an architecture cell.

### B. Latent Space Evolutionary Neural Architecture Search Algorithm

From the generative model of Transformer-based variational autoencoder encoding module, the optimization algorithm can be applied to search for the optimal architecture in the continuous latent space. The latent space is constructed based on the multivariate Gaussian distribution. Compared to the discrete search space, the continuous latent space gathers the promising architectures to the same region. The CMA-ES algorithm is chosen as the optimization algorithm to search for the optimal architecture due to its powerful versatility and applicability.

TABLE I
THE PERFORMANCE COMPARISON ON NAS-BENCH-101.

| Encoding methods | NAS methods | Queries | Avg. acc(%) |
|---|---|---|---|
| structure-aware encodings | BANANAS [14] | 200 | 94.09 |
| structure-aware encodings | NAO [15] | 1000 | 93.74 |
| Unsupervised computation-aware encodings | Arch2vec-BO [14] | 400 | 94.05 |
| Unsupervised computation-aware encodings | Arch2vec-RL [14] | 400 | 94.10 |
| Unsupervised computation-aware encodings | AG-Net [16] | 192 | 94.18 |
| Unsupervised computation-aware encodings | SAENAS-NE [17] | 150 | 94.08 |
| Zero-cost proxies | ProxyBO [18] | 150 | 94.04 |
| Zero-cost proxies | Synflow [19] | 150 | 91.68 |
| — | WeakNAS [20] | 200 | 94.18 |
| Hybrid computation-aware encodings | HEP-ENAS(ours) | 350 | 94.23 |

TABLE II
THE PERFORMANCE COMPARISON ON NAS-BENCH-201.

| NAS methods | Queries | CIFAR-10 | | CIFAR-100 | | ImageNet16-120 | |
|---|---|---|---|---|---|---|---|
| | | val (%) | test (%) | val (%) | test (%) | val (%) | test (%) |
| $\beta$–DARTS [21] | 11520 | 91.55 | 94.36 | 73.49 | 73.51 | 46.37 | 46.34 |
| PRE-NAS [22] | — | 91.37 | 94.04 | 71.95 | 72.02 | 45.16 | 45.34 |
| ProxyBO [18] | — | — | 91.46 | — | 73.48 | — | 47.18 |
| BANANAS [14] | 192 | 91.56 | 94.30 | 73.49 | 73.50 | 46.65 | 46.51 |
| Arch2vec-BO [14] | — | 91.41 | 94.18 | 73.35 | 73.37 | 46.34 | 46.27 |
| Arch2vec-RL [14] | — | 91.32 | 94.12 | 73.13 | 73.15 | 46.22 | 46.16 |
| GANAS [23] | 444 | — | 94.34 | — | 73.28 | — | 46.80 |
| AG-Net [16] | 192 | 91.60 | 94.37 | 73.49 | 73.51 | 46.64 | 46.43 |
| SAENAS-NE [17] | 100 | 91.58 | 94.34 | 73.46 | 73.46 | 46.59 | 46.36 |
| HEP-ENAS(ours) | 200 | 91.61 | 94.37 | 73.49 | 73.51 | 46.60 | 47.21 |
| Optimal | — | 91.61 | 94.37 | 73.49 | 73.51 | 46.77 | 47.31 |

TABLE III
THE ABLATION STUDIES ON NAS-BENCH-MARKS.

| Dimension size | NAS-Bench-101 | NAS-Bench-201 | | | | | |
|---|---|---|---|---|---|---|---|
| | test (%) | CIFAR-10 | | CIFAR-100 | | ImageNet16-120 | |
| | | val (%) | test (%) | val (%) | test (%) | val (%) | test (%) |
| 16 | 94.20 | 91.59 | 94.37 | 73.47 | 73.51 | 46.48 | 47.10 |
| 32 | 94.23 | 91.61 | 94.37 | 73.49 | 73.51 | 46.60 | 47.21 |
| 64 | 94.21 | 91.61 | 94.37 | 73.49 | 73.48 | 46.65 | 47.16 |

In the Algorithm 1, we use unsupervised learning to pre-train the Transformer-based variational autoencoder encoding module, which assists to pre-train the graph flow-based encoding module by supervised learning. When the pre-training of hybrid encoding-based performance predictor is completed, the CMA-ES algorithm is employed to explore the optimal architecture in the continuous latent space. During the search phase, the hybrid encoding-based performance predictor will be continuously updated until the evaluated architecture reaches the maximum evaluated number.

## III. EXPERIMENTS

To assess the effectiveness and performance of the proposed hybrid encoding-based predictor, the HEP-ENAS is evaluated on the widely used NAS-Benchmarks such as NAS-Bench-101 [12] and NAS-Bench-201 [13]. The NAS-Benchmarks commonly consist of the trained and evaluated architectures with related information, and offer a channel where we can immediately acquire the performance indicators of architectures without the time cost on training the architectures.

### A. Experiments on NAS-Benchmark-101

NAS-Benchmark-101 provides 423k unique architectures with the corresponding training information on CIFAR-10 classification task such as flops, parameters, accuracy and training time. NAS-Bench-101 builds the architecture by stacking cells and restricting the search space to a cell. The cells are defined by DAGs, where the nodes represents the operations and the adjacency matrix represents the connection of different operations. For each architecture in NAS-Bench-101, the maximum number of nodes is set as 7 and the maximum number of edges as 9, and it has different operations such as 3 × 3 convolution, 3 × 3 max-pooling and 1 × 1 convolution to choose.

As shown in Table I, we compare HEP-ENAS with several different encoding-based NAS methods such as structure-aware encodings, unsupervised computation-aware encodings and zero-cost proxies. It is evident that HEP-ENAS achieves the best result on NAS-Bench-101 with the average accuracy of 94.23%, which is the top 2 on NAS-Benchmark-101. The results show the effectiveness of the proposed hybrid

encoding-based predictor compared to other encoding-based NAS methods.

### B. Experiments on NAS-Benchmark-201

NAS-Benchmark-201 contains 15K architectures with their training, validation and test accuracy trained on CIFAR-10, CIFAR-100 and ImageNet-16-120 datasets. Each architecture is generated by 4 nodes and 6 operations. The NAS-Bench-201 provides zero, $3 \times 3$ convolution, skip connection, $1 \times 1$ convolution and $3 \times 3$ average pooling as candidate operations.

To further demonstrate the effectiveness of the proposed approach, the HEP-ENAS is compared with the encoding-based NAS methods and several other popular NAS methods on NAS-Benchmark-201 shown in Table II. The results show the proposed HEP-ENAS outperforms these methods.

### C. Ablation studies

The Transformer-based variational autoencoder encodings play an essential role in HEP-ENAS. The optimization algorithm search for the optimal neural architecture based on the Transformer-based variational autoencoder encodings, and then, the hybrid encoding-based performance predictor utilizes the Transformer-based variational autoencoder encodings to acquire the prediction scores. Thus, a series of ablation studies are conducted on the NAS-bench-marks to investigate the impact of different dimension sizes of Transformer-based variational autoencoder encodings on the final performance of HEP-ENAS. As shown in Talbe III, the dimension size with 32 which is chosen as the default in HEP-ENAS substantially achieves the optimal performance on NAS-benchmarks.

## IV. CONCLUSIONS

In this paper, the hybrid encoding-based performance predictor is developed to better balance the budget between efficiency and performance of the predictor-based NAS. The hybrid encoding-based performance predictor combines the unsupervised Transformer-based variational autoencoder encodings with the supervised graph flow-based encodings to better encode the architectures and improve the efficiency of downstream optimization algorithm. Furthermore, the CMA-ES is emploid to search for neural architectures in the continuous latent space with the hybrid encoding-based performance predictor to improve the search efficiency. The experiments on NAS-Benchmark-101 demonstrate the effectiveness of the proposed hybrid encoding-based predictor compared to other encoding-based NAS methods, and the experimental results on NAS-Benchmark-201 show competitive final performance of the proposed HEP-ENAS to several popular NAS methods.

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
