# OpenReview forum: "Evolutionary Neural Architecture Search with Performance Predictor Based on Hybrid Encodings"
_IEEE.org/ICIST/2024/Conference — IEEE ICIST 2024 Conference Submission_

### Official Review · Reviewer_UkKs · 2024-08-22
**Evolutionary Neural Architecture Search with Performance Predictor Based on Hybrid Encodings**

**Rating:** 7
**Confidence:** 5

**Review:**

In this paper,  a hybrid encoding-based predictor building upon two computation-aware encodings with different training approaches is proposed. This work is well organized and novel. Below are some comments.
(1)	The contributions should be illustrated in a clearer manner. For example, what is the main improvement of the paper compared to the existing results. The authors should explain the unique contributions of this paper.
(2)	The simulation results should be explained more carefully.
(3)	The paper is well presented, spelled correctly. I recommend authors to carefully read the entire paper to find possible misspellings.

---

### Official Review · Reviewer_tdoC · 2024-08-22
**Well-done work**

**Rating:** 6
**Confidence:** 3

**Review:**

The paper introduces a promising method for improving Neural Architecture Search through the use of a hybrid encoding-based performance predictor. This topic is interesting, the following comments need to further consider: 1. Provide a brief explanation of how the hybrid encoding-based predictor works, particularly how it combines unsupervised and supervised learning techniques. This would give readers a clearer understanding of the innovation behind the approach. 2. Incorporate some specific comparative results from the NAS benchmarks to illustrate the effectiveness of the proposed method. Highlighting key metrics where HEP-ENAS outperforms other methods would add weight to the claims made in the abstract. 3. Consider simplifying some of the technical jargon or providing brief explanations where necessary.

---

### Official Review · Reviewer_BkKr · 2024-08-23
**Generally good, but there are still some problems**

**Rating:** 8
**Confidence:** 5

**Review:**

The manuscript titled "Evolutionary Neural Architecture Search with Performance Predictor Based on Hybrid Encodings" presents a novel approach to Neural Architecture Search (NAS), which aims to improve the efficiency and effectiveness of finding optimal neural network architectures. The paper introduces a hybrid encoding-based performance predictor that combines unsupervised and supervised computation-aware encodings. The proposed hybrid encoding combines a Transformer-based variational autoencoder for unsupervised learning with a graph flow-based encoding module for supervised learning. This combination aims to enhance the encoding of neural architectures and improve the prediction accuracy of their performance.

Comments:

1.The hybrid encoding approach is innovative, but a clearer comparison with existing methods would better highlight its advantages.

2.Discuss potential limitations and how the method might perform in different scenarios or with various neural architectures.

3.Simplify the mathematical formulations and provide a glossary for better accessibility to a broader audience.

4.Check for spelling errors, particularly in technical terms.

---

### Decision · Program_Chairs · 2024-09-08

Accept (Oral)